# Mucosal and Serum Neutralization Immune Responses Elicited by COVID-19 mRNA Vaccination in Vaccinated and Breakthrough-Infection Individuals: A Longitudinal Study from Louisville Cohort

**DOI:** 10.3390/vaccines13060559

**Published:** 2025-05-24

**Authors:** Lalit Batra, Divyasha Saxena, Triparna Poddar, Maryam Zahin, Alok Amraotkar, Megan M. Bezold, Kathleen T. Kitterman, Kailyn A. Deitz, Amanda B. Lasnik, Rachel J. Keith, Aruni Bhatnagar, Maiying Kong, Jon D. Gabbard, William E. Severson, Kenneth E. Palmer

**Affiliations:** 1Center for Predictive Medicine for Biodefense and Emerging Infectious Diseases, University of Louisville, Louisville, KY 40202, USA; lalit.batra@louisville.edu (L.B.); divyasha.saxena@louisville.edu (D.S.); maryam.zahin@louisville.edu (M.Z.); megan.bezold@louisville.edu (M.M.B.); kathleen.kitterman@louisville.edu (K.T.K.); kailyn.deitz@louisville.edu (K.A.D.); amanda.lasnik@louisville.edu (A.B.L.); jon.gabbard@louisville.edu (J.D.G.); william.severson@louisville.edu (W.E.S.); 2Department of Bioinformatics and Biostatistics, School of Public Health and Information Sciences, University of Louisville, Louisville, KY 40202, USA; tri.poddar19@gmail.com (T.P.); maiying.kong@louisville.edu (M.K.); 3Christina Lee Brown Envirome Institute, School of Medicine, University of Louisville, Louisville, KY 40202, USA; alok.amraotkar@louisville.edu (A.A.); rachel.keith@louisville.edu (R.J.K.); aruni.bhatnagar@louisville.edu (A.B.); 4Department of Pharmacology and Toxicology, School of Medicine, University of Louisville, Louisville, KY 40202, USA; 5James Graham Brown Cancer Center, University of Louisville, Louisville, KY 40202, USA

**Keywords:** COVID-19, mRNA vaccine, antibodies, IgA and IgG, systemic response, mucosal response, breakthrough infections, Omicron, hybrid immunity, microneutralization (MN)

## Abstract

**Background/Objectives**: The COVID-19 pandemic, caused by severe acute respiratory syndrome coronavirus type-2 (SARS-CoV-2), has resulted in 777 million cases worldwide. Various vaccines have been approved to control the spread of COVID-19, with mRNA vaccines (Pfizer and Moderna) being widely used in the USA. We conducted a prospective longitudinal study to analyze the immune response elicited by two/three and four doses of monovalent mRNA vaccines in both vaccinated individuals and those who experienced breakthrough infections. Participants were stratified into different age groups: 18–40, 41–60, and over 60 years. **Methods**: We assessed cross-variant neutralization responses in two cohorts—Cohort I: n = 167 (serum), Cohort II: n = 92 (serum and nasal swab) samples—using infectious virus microneutralization assay (MN) and antibody (IgG or IgA) binding ELISA titers to the spike protein receptor binding domain (RBD). Samples were collected from the Louisville Metro–Jefferson County Co-Immunity Project, a federally funded, population-based study for the surveillance of SARS-CoV-2 in Jefferson County, Kentucky during 2020–2022, involving both health care workers and a local community. **Results**: Individuals who received two doses of the mRNA vaccine exhibited reduced neutralization against Beta, Delta, and Omicron BA.1 variants compared to wildtype Wuhan, with further decline observed six months post-booster vaccination. However, individuals who experienced natural COVID-19 infection (breakthrough) after receiving two vaccine doses showed enhanced neutralization and antibody responses, particularly against Omicron BA.1. Following the 3rd dose, antibodies and neutralization responses were restored. Among triple-vaccinated individuals, reduced neutralization was observed against Omicron variants BA.1, BA.5, and BA.2 compared to Wuhan. Neutralization responses were better against BA.2 variant compared to BA.1 and BA.5. However, individuals who received three doses of vaccine and experienced a breakthrough infection (n = 45) elicited significantly higher neutralizing antibodies responses against all Omicron subvariants compared to vaccinated individuals. Interestingly, nasal swab samples collected from volunteers with breakthrough infection showed significantly elevated spike-reactive mucosal IgA antibodies and enhanced cross neutralization against BA.1, BA.2, and BA.5 compared to individuals who received only three vaccine doses. **Conclusions**: mRNA vaccination elicits a strong systemic immune response by boosting serum neutralizing antibodies (NAb), although this protection wanes over time, allowing new variants to escape neutralization. Breakthrough individuals have extra enrichment in nasal NAb offering protection against emerging variants. This longitudinal immune profiling underscores the strengthening of pandemic preparedness and supports the development of durable mucosal vaccines against respiratory infectious disease.

## 1. Introduction

The COVID-19 pandemic has led to substantial mortality and morbidities around the globe [1,2]. Several risk factors contribute to increased disease severity in adults, including advanced age, male sex, pre-existing comorbidities, and racial/ethnic disparities [2]. Additionally, elevated levels of pro-inflammatory cytokine and disease-related complications have been associated with severe outcomes and mortality in COVID-19 patients [1,3,4]. Most COVID-19-infected individuals present with respiratory symptoms such as fever, chills, cough, and difficulty breathing [5,6]. Severe cases can lead to respiratory failure, necessitating ventilatory support and causing multi-organ damage [7]. Due to the rapid spread and global burden of the disease, researchers and pharmaceutical companies swiftly developed vaccines using both established and novel technologies.

To date, multiple COVID-19 vaccines have been authorized worldwide, including Pfizer-BioNTech, Moderna, Janssen/Johnson & Johnson, AstraZeneca/Oxford, and Novavax. These vaccines have demonstrated safety and efficacy in preventing severe illness caused by SARS-CoV-2. In the U.S., the primary vaccines administered were mRNA-based Pfizer and Moderna vaccines targeting the SARS-CoV-2 spike protein, and adenovirus-based Johnson & Johnson vaccine [8]. For the 2024–2025 COVID-19 vaccination campaign, the authorized vaccines for individuals aged 12 years and older include the Pfizer-BioNTech and Moderna vaccines, as well as a protein subunit vaccine by Novavax [9].

Worldwide, 13.6 billion doses of COVID-19 vaccines have been administered to reduce subsequent SARS-CoV-2 infections. However, breakthrough infections have been reported following the emergence of Omicron subvariants due to immune escape mutations [10]. The occurrence of breakthrough infections may be influenced by several factors, including viral characteristics, immune responses, host determinants, and vaccine properties. Determining the true rate of breakthrough infections requires population-based studies on vaccine effectiveness, as well as neutralizing antibody assessments as a surrogate measure of immune protection [10].

Omicron, which was first identified in South Africa in November 2021, rapidly became the dominant variant and further evolved into various sublineages such as BA.1 and BA.2, containing at least 37 spike protein mutations accounting for their high transmissibility and antibody evasion properties [11]. BA.2 further evolved into sublineages BA.2.12.1, BA.2.75, and BA.2.75.2, which collectively accounted for 0.8% of total SARS-CoV-2 infections in the United States during November 2022 [12]. Subsequently, BA.2 Omicron clade rapidly evolved into BA.4 and BA.5 with further sublineages such as BF.7, BQ.1, and BQ.1.1 accounting for 4.4%, 7.8%, 25.5%, and 24.2% of total cases in United States during January 2022–May 2023, respectively [12]. During summer and fall 2023, BA.5-derived sublineage XBB.1 and multiple descendants of XBB with immune escape substitutions emerged and reached >10% prevalence, including EG.5, FL.1.5.1, HV.1, and JN.1 variants [13]. As a result, there is a continued interest in developing next-generation vaccines capable of eliciting broader and longer-lasting immunity against emerging variants [14].

Due to multiple mutations and relatively lower neutralization against BA.5 subvariants, booster doses have been shown to enhance protection for a limited duration, though they do not elicit a strong mucosal immune response, which is critical for preventing infection at the virus’s entry site such as respiratory tract [15], oral mucosa [15], and conjunctiva [16]. Recent studies analyzing neutralizing antibodies in healthcare workers [17] and in a cohort of general populations of Uganda [18] reported that more than half of individuals who received three vaccine doses still became infected with SARS-CoV-2 Omicron variants, 2–3 months post booster administration.

With increasing numbers of breakthrough infections, the need to determine the breadth and durability of circulating and mucosal neutralizing antibodies is of utmost clinical importance, as current mRNA vaccines do not provide mucosal immunity. Mucosal IgA antibodies generated by natural SARS-CoV-2 infection have been shown to provide protection [19,20] against reinfections with Omicron subvariants for up to 7 months [21]. Furthermore, it is evident that significant numbers of individuals contracted COVID-19 during the Omicron wave despite receiving two or three doses of an mRNA vaccine, emphasizing the need for further assessment of their antibody responses [22].

To address these questions, we characterized the neutralization profile of sera and nasal swab samples collected from individuals who were either vaccinated alone or experienced breakthrough infections in a study conducted during 2020–2022. The study outcome provided a valuable insight into the roles of mucosal and systemic immunity in vaccine effectiveness and hybrid immunity.

## 2. Materials and Methods

### 2.1. Study Design and Participants

The study design consists of two cohorts. Cohort I (n = 167) includes blood samples from individuals living in Jefferson County, Kentucky who have received two doses of an mRNA vaccine (Pfizer or Moderna). Though samples were collected over 8 waves between June 2020 and August 2021 using random (n = 7296) and volunteer (n = 7919) community-wide sampling as published earlier [23], only samples from 2021 are included in this study, as the mRNA vaccine was not available before then. Informed consent was obtained electronically from each participant, as well as self-reported information on demographics, health, lifestyle, sex, age, and COVID-19 vaccination status. The objective of this part of the study was to investigate the cross-variant neutralization capabilities of human sera collected from vaccinated individuals of different aged groups, both within (n = 75) or after 6 months (n = 75) of receiving the second vaccine dose. Neutralization of Wuhan, Beta, Delta, and Omicron BA.1 variants was examined. Additionally, Cohort I was used to compare the neutralization response between vaccinated-only individuals and those who experienced breakthrough (n = 17) infections (hybrid response). Cohort II samples were collected in July 2022 from 92 individuals associated with the University of Louisville. Because a booster vaccination was encouraged by this time, Cohort II was comprised of individuals who had received three (n = 81) or four (n = 11) doses of a COVID-19 mRNA vaccine (Pfizer or Moderna). This part of the study was designed to evaluate the neutralizing-antibody (IgG and IgA) responses from both sera and nasal swab against wild type and mutant VOCs such as Omicron BA.1, BA.2, and BA.5. The objective was to compare the immune responses of triple-vaccinated (n = 40) individuals to those who had breakthrough infection (n = 41) during the Omicron dominance period. The neutralizing capabilities of serum and nasal swab samples were characterized using microneutralization assay (MN) and enzyme-linked immunosorbent assay (ELISA) against receptor-binding domain (RBD) protein.

### 2.2. Participant Inclusion and Exclusion Criteria

For Cohort I, participants (n = 167) were selected from random (n = 7296) and volunteer (n = 7919)-collected sera sample based on various inclusion and exclusion criteria. The inclusion criteria were two-dose (n = 150) monovalent mRNA vaccination (Pfizer or Moderna) or two-dose vaccinated and breakthrough infections (n = 17). The groups were further stratified into age groups 18–40, 41–60, and over 60 years, with equal participants in each group. Effort was made to have an equal proportion of male/female; Pfizer/Moderna; as well as all the population races from Louisville Metro, including White/Middle Eastern/North African/Asian/Black/African American/Hispanic, and Latino. The exclusion criteria were underlying health conditions and comorbidities including chronic health conditions such as heart disease, high blood pressure, COPD, autoimmune disease, HIV/AIDS, cancer, thyroid disease, liver disease, chronic kidney disease. Individuals were non-smokers, as those who used cigarettes, E-cigarettes, cigarillos, or vapes were excluded from the study. The selected individuals were not at the time of sample collection taking any medications including antivirals, ace inhibitors, and immunosuppression medications. For Cohort II (n = 92), individuals had to have received three doses (n = 40) or four doses (n = 11) of a monovalent mRNA vaccination (Pfizer or Moderna) or three vaccine doses and experienced breakthrough infections (n = 41). All the other inclusion and exclusion criteria for Cohort II were similar as described for Cohort I selection.

### 2.3. Human Samples and Serology

Trained medical staff collected nasal swab and finger-prick blood samples. The nasal swabs were collected dry and then eluted in PBS (1 mL) and stored at −80 °C until use. Blood samples were centrifuged, and sera were collected, incubated at 56 °C for 30 min for heat inactivation, and then stored at −80 °C until use.

### 2.4. Cells and Viruses

TMPRSS2-overexpressing VeroE6 recombinant cells (BPS Biosciences, San Diego, CA, USA, Catalog #78081) were used for viral infection and propagation, avoiding cell culture adaptation-mutations which may arise during culture in VeroE6 cells. The cells were maintained in Dulbecco’s Modified Eagle Medium (DMEM, Thermo Fisher Scientific, Waltham, MA, USA, Cat# 11995-065) supplemented with 2 mM Glutamine (Thermo Fisher Scientific, Waltham, MA, USA, Cat# 25030081), 100 U/mL penicillin/streptomycin (Thermo Fisher Scientific, Waltham, MA, USA, Cat# 15070063) and 10% FBS (Sigma-Aldrich, St. Louis, MO, USA, Cat# F4135). All viral culture and infectious virus assays were conducted in the Biosafety Level 3 containment facility at the University of Louisville (IBC #20-271; “Antiviral Discovery of Small Molecules and Antiviral Targets of Coronaviruses”). SARS-CoV-2 strains used for assays included the following: Wuhan (nCoV/Washington/1/2020) virus was from Biodefense and Emerging Infections Research Resources Repository (#NR-52281, “The following reagent was deposited by the Centers for Disease Control and Prevention and obtained through BEI Resources, NIAID, NIH: SARS-Related Coronavirus 2, Isolate hCoV-19/USA-WA1/2020, NR-52281”), Delta (B.1.617.2) from World Reference Center for Emerging Viruses and Arboviruses (#GNL-1205), and Beta (B.1.351) from BEI resources (#NR-555282, “The following reagent was obtained through BEI Resources, NIAID, NIH: SARS-Related Coronavirus 2, Isolate hCoV-19/USA/MD-HP01542/2021 (Lineage B.1.351), in *Homo sapiens* Lung Adenocarcinoma (Calu-3) Cells, NR-55282, contributed by Andrew S. Pekosz”), BA.1 (# NR-56461, “The following reagent was obtained through BEI Resources, NIAID, NIH: SARS-Related Coronavirus 2, Isolate hCoV-19/USA/MD-HP20874/2021 (Lineage B.1.1.529; Omicron Variant), NR-56461, contributed by Andrew S. Pekosz”), BA.2 (#NR-56781, “The following reagent was deposited by the Centers for Disease Control and Prevention and obtained through BEI Resources, NIAID, NIH: SARS-Related Coronavirus 2, Isolate hCoV-19/USA/NY-MSHSPSP-PV56475/2022 (Lineage BA.2.12.1; Omicron Variant), NR-56781, deposited by Dr. Viviana Simon”), BA.5 (#NR-56798, “The following reagent was obtained through BEI Resources, NIAID, NIH: SARS-Related Coronavirus 2, Isolate hCoV-19/South Africa/CERI-KRISP-K040013/2022 (Lineage BA.5; Omicron Variant), NR-56798, deposited by Dr. Alex Sigal”).

### 2.5. Infectious Virus Microneutralization Assay

Microneutralization assays were performed as described previously with modifications [24]. Briefly, serum/swab specimens were two-fold serially diluted starting from 1:5 (n = 3 technical replicates per serum/swab specimens per virus strain). Each dilution was mixed with 100 TCID_50_/mL of different SARS-CoV-2 VOCs for 1 h to allow antigen-specific antibodies to bind to the virus, after which the mixture was added to VeroE6/TMPRSS2 cell monolayers in 96-well plates. Following incubation for 3 days (Wuhan, Beta, Delta) or 4 days (Omicron BA.1, BA.2 and BA.5 variant), plates were observed under an inverted light microscope, and cytopathic effect (CPE) was visually scored for each well (5, 4, 3, 2, 1, and 0) by two independent observers. A well was assigned a score of 5 or 100% viable cell if no CPE was present and score of 0 or 100% viral infected cells if full CPE was observed. The results were plotted using 5-parameter dose–response curve (non-linear regression curves [log(inhibitor) vs. normalized response—Variable slope] in GraphPad Prism software version 9.1.1. The inhibitory dose 50 (ID_50_) also referred to as 50% inhibitory dose or viral neutralization 50 titers (VNT50), was determined as reciprocal of highest serum dilution that protected more than 50% of the monolayer cells from CPE and was reported as the geometric mean titer (GMT) of triplicates.

### 2.6. Anti-RBD Assay for Wild Type and Variants of Concerns (VOCs)

Enzyme immunoassay for anti-RBD IgG antibody was performed as described previously with modifications [24]. Spike RBD proteins were purchased from Immune Technology Corp, New York, NY, USA (SARS-CoV-2 Wuhan, His-tagged, Cat# IT-002-036p) and Sino Biological, Wayne, PA, USA [SARS-CoV-2 Delta Spike RBD (L452R, T478K) Protein, Cat# 40592-V08H90; SARS-CoV-2 Beta Spike RBD (K417N, E484K, N501Y) Protein, Cat# 40592-V08H85; SARS-CoV-2 BA.1 (Omicron) Spike RBD Protein, Cat# 40592-V08H121; SARS-CoV-2 BA.2 (Omicron) Spike RBD Protein, Cat# 40592-V08H123; SARS-CoV-2 BA.5 (Omicron) Spike RBD Protein, Cat# 40592-V08H131].

Briefly, 96-well immunoplates (Thermo Fisher Scientific, Waltham, MA, USA, Cat#442404) were coated with 100 μL/well (0.2 μg/well) of His-tagged SARS-CoV-2 spike RBD in PBS (pH 9.6) overnight at 4° C, followed by incubation with a blocking reagent (Thermo Fisher Scientific, Waltham, MA, USA, Cat#37572) for 1 h at room temperature. 100 μL of heat-inactivated serum or nasal swab samples at two-fold dilutions (starting from 1:100 for serum and 1:5 for swab) were added to the wells and incubated at 37° C for 30 min. The attached human antibodies were detected using horse radish peroxidase-conjugated Rabbit anti-Human IgG (Thermo Fisher Scientific, Waltham, MA, USA, Cat# PA1-28587) from serum samples or Goat anti-Human IgA (Southern Biotech, Birmingham, AL, USA, Cat# 2050-05) from nasal swabs samples. The reaction was developed by adding 100 μL of 3,3′,5,5′-tetramethylbenzidine single solution (Thermo Fisher Scientific, Waltham, MA, USA, Cat# 34029) and stopped after 5 min with 100 μL of 1N Hydrochloric Acid Solution (Honeywell Research Chemicals, Charlotte, NC, USA, Cat#35328-1L). The optical density (OD) was read at 450 nm (BioTek Synergy LX Multimode Plate Reader, Agilent Technologies, CA, USA) For normalization, naïve human sera (pre-COVID sera, Sigma Aldrich, St. Louis, MO, USA, Cat# H4522) were used at 1:100 dilution. Endpoint IgG titers against SARS-CoV-2 variant specific RBD were determined based on twice the absorbance cutoff value of naïve sera at 1:100 dilution. Endpoint IgA binding titers against SARS-CoV-2 variant specific RBD were determined based on twice the absorbance cutoff value of plate background. Serum and nasal swab samples endpoint titers were reported as geometric mean titers of replicates.

### 2.7. Statistics

Samples from each participant were cultured with different COVID-19 viruses. The two primary endpoints, the 50% inhibitory dilutions and RBD IgG levels, were measured under each virus culture condition. Descriptive statistics, including geometric mean and standard deviation, along with graphical presentations, were first used to summarize the observed data. Rigorous statistical inference was then conducted using a linear mixed-effects model for each endpoint in each cohort. Fixed effects included the main effects of virus type, age group, and vaccination period, while random effects accounted for individual participants. Two-way interactions between the main effects were included to assess whether the effects of virus type depended on age group and vaccination period. Additionally, Wald tests for different contrasts were used to examine group differences. The result was considered statistically significant if the *p*-value was less than 0.05. All statistical analyses were performed using the statistical analysis software R Version 4.3.0 (https://www.r-project.org/).

## 3. Results

### 3.1. Reduced Sensitivity of SARS-CoV-2 Variants Beta, Delta, and Omicron BA.1 to Antibody Neutralization from Sera of Pfizer- or Moderna-Vaccinated Individuals

We tested the vaccine-elicited antibodies neutralization capabilities against SARS-CoV-2 VOCs including Beta, Delta, and Omicron BA.1 from sera samples of two-dose vaccinated Pfizer- or Moderna-vaccinated individuals (Figure 1). The sera samples were collected within 6 months of the last vaccine dose and were from different age groups viz.18–40 (n = 25), 41–60 (n = 25), and >60 (n = 25). We observed the highest neutralization potential in the 18–40 age group (Figure 1A) against Wuhan (mean ID50; 9881), followed by Delta (mean ID50; 1285), Beta (mean ID50; 767), and least for Omicron BA.1 (mean ID50; 146). For Age Group 41–60 (Figure 1B), we observed slightly less neutralization potential than in the 18–40 age group, and mean ID50 values were 5055 for Wuhan, 1072 for Delta, 751 for Beta, and 133 for Omicron BA.1. For Age Group > 60 (Figure 1C), we observed the least neutralization and mean ID50 values were in the range of 4224 for Wuhan, 620 for Delta, 194 for Beta, and 49 for Omicron BA.1. The differences in terms of neutralization titers for parental Wuhan strain compared to other VOCs were found to be significant in two-dose vaccinated (within 6 months) different age group individuals.

To study the duration of vaccine induced humoral immunity, we also tested the sera samples from individuals who were beyond 6 months post receiving a second vaccine dose with similarly assigned age groups. We observed the moderate neutralization potential (Figure 1A) of 18–40 age group (n = 25) against Wuhan (mean ID50; 3226), followed by Delta (mean ID50; 582), Beta (mean ID50; 206), and least for Omicron BA.1 (mean ID50; 90). For Age Group 41–60 (n = 26), we observed slightly less neutralization potential (Figure 1B), and mean ID 50 values were in the range of 2701 for Wuhan, 401 for Delta, 133 for Beta, and 42 for Omicron BA.1. For Age Group > 60 (n = 24), we observed relatively less neutralization potential (Figure 1C), and mean ID50 values were in the range of 1960 for Wuhan, 402 for Delta, 99 for Beta and 35 for Omicron BA.1. The differences in terms of neutralization titers for parental Wuhan strain compared to other VOCs were found to be significant in two-dose vaccinated (over 6 months) different age group individuals.

With all age groups, more neutralization was induced for all SARS-CoV-2 VOCs when the post-vaccination duration was less than 6 months, compared to more than 6 months. The significant differences among all VOCs for different virus types and vaccine duration have been summarized in Appendix A. Individuals assigned to different age groups were significantly different from each other, and we observed the highest virus neutralization in the 18–40 vaccinated age group. The table depicting the virus neutralization from different age groups (combined under and over 6 months duration) is shown in Appendix A.

Additionally, using commercial spike RBD proteins, we tested the IgG binding titers against SARS-CoV-2 VOCs, and similar patterns of results were observed in both vaccine duration and age group comparison setting (Figure 2). However, IgG binding titers did not differ significantly (*p* > 0.19) between the Beta and Delta variants for any age group with vaccine duration groups combined. Furthermore, age groups 18–40 and 41–60 showed a significant difference for the Wuhan virus but not for the other three virus types. The table summarizing significant differences between the vaccine duration and age groups is shown in Appendix A.

### 3.2. Increased Efficiency for Virus Microneutralization and IgG Binding to RBD of Different VOCs from Sera Samples Collected from Two-Dose Vaccinated and SARS-CoV-2 Infected (Breakthrough) Individuals

To observe the difference between natural immunity and vaccination-acquired immunity, we also tested sera (n = 17) from vaccinated only and SARS-CoV-2 infected (breakthrough) individuals. We observed improved SARS-CoV-2 VOCs virus neutralization (Figure 3A) and mean ID50 values were found to be in the range of 10,586 for Wuhan, 2897 for Beta, 4897 for Delta, and 673 for Omicron BA.1. The fold reduction in mean ID50 values in comparison to ancestral Wuhan strain was highest for Wuhan vs. Omicron (85.66), low for Wuhan vs. Beta (21.71), and least for Wuhan vs. Delta (6.78). We observed a similar pattern for IgG binding titers against SARS-CoV-2 VOCs specific commercial Spike RBD protein (Figure 3B) with fold reduction highest for Wuhan vs. Omicron (17.67), low for Wuhan vs. Beta (9.97), and least for Wuhan vs. Delta (3.79).

To compare the neutralization efficacies between natural infection versus vaccination, we compared neutralization titers of individuals in the highest antibody neutralization age group (n = 25), i.e., 18–40 (under 6 months) versus titers from natural infection group (n = 17) and observed significantly elevated neutralization for Beta (*p* < 0.017) and Delta (*p* < 0.016) following SARS-CoV infection. The results depicting the *p* value difference in terms of neutralization titers and IgG titers between infected versus vaccinated individuals have been summarized in Table 1. Interestingly, IgG titers against Spike RBD protein were not significantly increased after breakthrough infection (Table 2).

### 3.3. Reduced Sensitivity of New SARS-CoV-2 Omicron Variants BA.1 and BA.5, but Not BA.2 to Antibody Neutralization from Sera of Pfizer or Moderna Triple-Vaccinated Individuals

Next, we tested the neutralization potential of single booster vaccine-elicited antibodies against newly emerged SARS-CoV-2 Omicron Variants BA.1, BA.2, and BA.5 using virus microneutralization assay and RBD binding ELISA assays. Sera samples were collected post 6 months following third vaccine dose from different age groups individuals viz.18–40 (n = 20), 41–60 (n = 15), and >60 (n = 5). We observed significantly reduced virus neutralization against BA.1, BA.2, and BA.5 as compared with ancestral Wuhan strain (Figure 4A). Interestingly, boosted sera have better neutralization efficacies for the BA.2 variant compared to BA.1 and BA.5. The significant differences in reduction of neutralization titers for Omicron variants BA.1, BA.2, and BA.5 compared to parental Wuhan strain are shown in Figure 4A. The table comparing *p* values for neutralization efficacies (ID50) of different age group individuals against SARS-CoV-2 VOCs have been summarized in Appendix A. Similar patterns of the results were observed with IgG titers against SARS-CoV-2 VOCs specific commercial Spike RBD protein as shown in Figure 4C and Appendix A.

### 3.4. Higher Neutralization Potential Against New SARS-CoV-2 Omicron Variants BA.1, BA.2, and BA.5, from Sera of Pfizer or Moderna Triple-Vaccinated Individuals and SARS-CoV-2 Infected (Breakthrough) Individuals

To observe the neutralization potential from three-dose-vaccinated and SARS-CoV-2 infected individuals, we performed the microneutralization (Figure 4B) and ELISA based assays (Figure 4D) against Omicron sublineages using sera collected from individuals who received three doses of mRNA vaccine and subsequently contracted SARS-CoV-2 infection in between 2nd and 3rd dose. We observed enhanced infectious virus neutralization against BA.1, BA.2, and BA.5 across all combined age groups individuals (n = 41) viz.18–40 (n = 14), 41–60 (n = 21) and >60 (n = 6). In terms of ID50, we observed significant fold reduction (*p* < 0.5) for Wuhan vs. BA.1, whereas fold reduction was non-significant for Wuhan vs. BA.2 and Wuhan vs. BA.5 (Figure 4B). We did not observe any significant fold reduction for Wuhan vs. BA.1, BA.2 and BA.5 in terms of RBD binding IgG titers (Figure 4D). No significant difference was observed among the different age groups. The table comparing *p* values of different aged groups of people against SARS-CoV-2 variants have been summarized in Appendix A (ID50) and Appendix A (IgG titers).

More interestingly, we compared the neutralization efficacies between triple-vaccinated-with-infection versus triple-vaccinated-only individuals and observed significantly increased neutralization from breakthrough samples in terms of ID50 across all virus types (Table 3) as well as IgG endpoint titers against all viruses except Wuhan (Table 4).

### 3.5. Mucosal IgA Generated Against SARS-CoV-2 Omicron Infection Leading to Enhanced, Effective, and Cross Neutralization Against BA.1, BA.2, and BA.5 from Nasal Swab Samples of Triple Vaccinated and SARS-CoV-2 Infected Individuals

To decipher the role of mucosal immunity following SARS-CoV-2 infection, we assessed the nasal swabs from triple-vaccinated versus triple-vaccinated and omicron-infected individuals for microneutralization and IgA ELISA against SARS-CoV-2 VOCs specific commercial Spike RBD protein. We observed negligible virus neutralization (Figure 5A) and low mucosal IgA titers against SARS-CoV-2 specific RBD proteins (Figure 5C) from triple vaccinated individuals. However, we observed effective and enhanced virus neutralization (Figure 5B) from triple-vaccinated and Omicron-infected individuals against Wuhan followed by BA.2/BA.5 and least for BA.1. Most of the samples were below threshold limit for BA.1 virus neutralization. In terms of ID50, the observed fold difference in reduction was 5.24 for Wuhan vs. BA.1 (*p* < 0.001), 1.21 for Wuhan vs. BA.2 (*p* = 0.54), and 1.21 for Wuhan vs. BA.5 (*p* = 0.51). We also observed better IgA response from vaccinated and infected individuals compared to vaccinated-only individuals, and the observed fold of reduction was 7.67 for Wuhan vs. BA.1 (*p* < 0.001), 8.65 for Wuhan vs. BA.2 (*p* < 0.001), and 7.54 for Wuhan vs. BA.5 (*p* < 0.001) (Figure 5D). The *p* values for mucosal response between vaccinated and infected vs. vaccinated-only individuals was found to be significant both for ID50 (Table 5) and IgA titers against RBD proteins (Table 6).

Nasal Swabs (n = 40) were collected from individuals in different age groups—18–40 (n = 20), 41–60 (n = 15) and >60 (n = 5)—who received three doses of vaccine only (Figure 5A) and compared to nasal swabs (n = 41) collected from individuals in different age groups—18–40 (n = 14), 41–60 (n = 21) and >60 (n = 6)—who received three doses of vaccine and were later infected with Omicron (Figure 5B) COVID-19 (breakthrough infections). These samples were analyzed by live-virus neutralization assays against the ancestral Wuhan strain and the recently emerged Omicron sublineages BA.1, BA.2, and BA.5. The fold reduction, GMTs and significant difference in virus neutralization to BA.1, BA.2, and BA.5 relative to ancestral Wuhan strain are shown in Figure 5 and comparisons between ID50 values for infected vs. vaccinated individuals are listed in Table 5. The swabs were tested in duplicates and in each panel, data are mean from 2 independent experiments (* *p* < 0.05, ** *p* < 0.01, *** *p* < 0.001). For Figure 5C,D, IgA binding ELISA was conducted against RBD proteins of ancestral Wuhan strain or Omicron sublineages BA.1, BA.2, and BA.5. The fold reduction, GMTs, and significant difference in IgA binding titers against RBD proteins of BA.1, BA.2, and BA.5 relative to ancestral Wuhan strain are shown and comparisons between IgA titers of infected vs. vaccinated individuals are listed in Table 6.

### 3.6. Reduced Sensitivity of BA.1, BA.2, and BA.5 to Antibody Neutralization from Sera and Nasal Swabs of Double-Boosted (Quadruple-Vaccinated) Individuals

To determine if double booster vaccine-elicited antibodies neutralized newly emerged SARS-CoV-2 Omicron Variants BA.1, BA.2, and BA.5 we performed assays as stated previously for single boosted individuals. Sera samples were collected from individuals (aged > 60 years; n = 11) within 6 months of receiving a double booster dose of either the Pfizer or Moderna vaccines. We observed significantly reduced virus neutralization (*p* < 0.01) against BA.1, BA.2, and BA.5 as compared with the ancestral Wuhan strain in terms of ID50 (Figure 6A) with fold difference of 5.9 for Wuhan vs. BA.1, 4.3 for Wuhan vs. BA.2 and 7.1 for Wuhan vs. BA.2. Interestingly, we observed better neutralization efficacies against BA.2 compared to BA.1 (*p* = 0.097) or BA.5 (*p* = 0.003). We observed reduced RBD IgG endpoint titers (Figure 6B) against BA.1, BA.2, and BA.5 compared to the Wuhan strain with significant and fold differences of 7.2 for Wuhan vs. BA.1 (*p* = 0.020), 2.64 for Wuhan vs. BA.2 (*p* = 0.042) and 9.79 for Wuhan vs. BA.5 (*p* = 0.019).

In nasal swab samples collected from double boosted individuals, we observed negligible or very low virus neutralization (Figure 6C) against Wuhan, BA.1, BA.2, and BA.5 with most samples below the threshold limit. However, RBD IgA titers (Figure 6D) against Wuhan, BA.1, BA.2, and BA.5 could be determined. The observed fold difference was 2.76 for Wuhan vs. BA.1 (*p* < 0.01), 1.82 for Wuhan vs. BA.2 (*p* = 0.03) and 1.94 for Wuhan vs. BA.5 (*p* < 0.05).

## 4. Discussion

Since the emergence of SARS-CoV-2, the virus has been responsible for approximately 778 million confirmed cases, with 7.1 million deaths worldwide. Despite that fact, widespread vaccination efforts, including the administration of multiple doses of both monovalent and bivalent vaccines, new SARS-CoV-2 variants continue to evolve globally. Among these, Omicron subvariants such as BA.1, BA.2, BA.4/5, and the more recently emerged BA.2.10 recombinant subvariants XBB.1.5, XBB.1.16, JN.1, KP.2, KP.3, and LB.1 have been identified with increased mutations. These subvariants have shown high resistance to all six clinically available monoclonal antibodies used for SARS-CoV-2 neutralization, treatments, and vaccinations [12,13,25,26,27].

Our study reports the robust serum-neutralizing hybrid immunity arising from a combination of vaccination and natural infection. We further showed mucosal immunity (IgA) induced by hybrid immunity generates detectable IgA titers and cross-variant neutralization response against the Omicron subvariants viz., BA.1, BA.2, and BA.5. Sera collected either within 6 months or post 6 months of last dose following two doses of mRNA vaccination provided adequate neutralization response against the ancestral Wuhan strain but showed reduced activity against the Beta variant, followed by the Delta variant with negligible neutralization observed against the Omicron BA.1 variant. These findings are consistent with preliminary reports from other groups [28,29,30]. Interestingly, we observed that the Delta variant exhibited greater resistance to neutralizing antibodies (NAbs) than the Beta variant in samples collected within 6 months of 2nd vaccine dose, probably due to shorter life span and lower frequency of NAb-producing plasma cells [31].

Our studies further indicated that humoral response waned over time, as observed by low neutralization response toward the ancestral Wuhan strain as well as Beta, Delta, and Omicron BA.1 variants, in serum collected post 6 months of the 2nd mRNA vaccine dose. Previous studies also reported that compared to Wuhan strain, neutralizing antibodies which persist post 6 months of vaccination, neutralizing antibodies against Omicron BA.1 from 2-dose-immunized individuals belonging to different aged groups were decreased over 100 fold in post 6 month of vaccination compared to within 6 months of vaccination [31,32]. This decline in neutralizing antibody levels was accompanied by a significant reduction in protection, with immunity waning considerably after 6 months. It has also been reported that the total anti SARS-CoV-2 antibodies levels declined gradually between 3–6 months following 2-dose mRNA vaccination, and further diminished post 6 months [33]. Furthermore, two doses of mRNA vaccination were not effective in preventing Omicron BA.1 infection [34]. Multiple studies have reported low Omicron BA.1 neutralization response as compared to ancestral Wuhan Strain with further substantial reduction in neutralization between 28 days and 6 months after the second dose [34,35]. These findings support the Centers for Disease Control (CDC) recommendations for a booster dose after 5 months of mRNA vaccination to maintain adequate immune protection.

Next, we sought to determine the effect of aging on antibody neutralization responses following two doses of mRNA vaccination by examining the responses of individuals in three groups. viz., 18–40 (young), 41–60 (intermediate), and >60 years (older). We observed the highest neutralization response towards multiple SARS-CoV-2 VOCs in the younger age group, both vaccinated within 6 months, and post 6 months of their last mRNA vaccine dose. Multiple studies have indicated that older individuals exhibited weaker humoral responses compared to individuals in younger and intermediate age groups, with antibody-mediated neutralization in plasma declining universally over time [36,37]. Thus, older adults may remain at high-risk as a population prone to COVID-19 infection despite vaccination, which warrants a need for frequent booster doses [37].

However, we observed improved and effective neutralization of the Beta, Delta, and Omicron BA.1 variants in the group of individuals who experienced SARS-CoV-2 breakthrough infection after two vaccine doses. This finding suggests that breakthrough infections serve as an effective immune booster, reinforcing the benefit for booster vaccination for sustained vaccine effectiveness [32,34]. Previous studies reported that infection-acquired immunity remained active for more than a year [38]. After breakthrough infection, the observed fold reduction in the NT_50_ values for Omicron BA.1 (GMT = 673) was in the range of 85x, as compared to the ancestral Wuhan strain (GMT = 1586). This clearly indicates breakthrough infections further improved the magnitude and breadth of SARS-CoV-2 neutralizing antibody responses [39]. The Omicron variants exhibit increased mutations and greater transmissibility than other SARS-CoV-2 variants, and it has been further proven that infection with Omicron variant prevents re-infection with other Omicron subvariants more efficiently than infection with non-Omicron variants [40].

Several studies have reported poor neutralizing efficacy against different SARS-CoV-2 VOCs following two doses of mRNA vaccine, whereas natural SARS-CoV-2 infection or administration of a third vaccine dose strongly boosts neutralization responses [29,38,39]. Interestingly, when the second dose of vaccine was administered with an extended interval, higher neutralization responses were observed after the second dose [41], and no further increase was noted after the third dose [42]. Hence, there is no clear documentation that additional doses of the original SARS-CoV-2 vaccines after the second or third dose will result in increased responses against VOCs.

We observed better neutralization responses against BA.1, BA.2, and BA.5 variants following the 3rd mRNA dose. Despite the significant increase in neutralization after the third vaccine dose, fold reduction in the GMTs against BA.1., BA.2, and BA.5 were 4.12-, 2.48-, and 12.34-fold, respectively, relative to the ancestral Wuhan strain. These results support the findings that BA.5 exhibits the greatest escape from vaccine-elicited neutralization among the Omicron subvariants [43]. In contrast, we observed higher neutralization against BA.5 and other Omicron sublineages in individuals who received three doses of vaccine and had natural COVID-19 infection. Sera from these individuals (n = 41) showed higher GMTs of 1277, 1985, and 1633 against BA.1, BA.2, and BA.5, respectively. These results suggest that natural infection, when combined with vaccination, is more effective than additional vaccine doses in boosting the magnitude and breadth of neutralization against all Omicron sublineages, especially against BA.5. Based on infection date, we predicted that individuals in Cohort II were likely infected with Omicron subvariants. As reported earlier, BA.1 infection enhanced neutralization efficacy, and this was fully dependent on previous vaccination, as BA.1 infection alone failed to elicit greater neutralization against Omicron sublineages in unvaccinated people [44]. Several studies have shown the immunity elicited by BA.1 infection offers limited protection against BA.4/BA.5, which have the strongest selective advantage in evading neutralization, particularly in unvaccinated or BA.1-infected individuals [45,46]. Given the closer genetic relationship between Omicron BA.2 and BA.4/BA.5 compared to BA.1, we deduced that our samples were mainly from BA.2 breakthrough infections which likely have shifted the cross-neutralization activity towards BA.2 and BA.4/BA.5.

The findings of our study are in line with public vaccine efficacy reports analyzing thousands of individuals across different countries [47,48]. For example, the effectiveness of three doses of Pfizer mRNA was reported as 52% in individuals without prior infection and 77% in those with prior SARS-CoV-2 infection, underscoring the added benefit of hybrid immunity [48]. Prior studies have shown that hybrid immunity due to vaccination and SARS-CoV-2 infection confers strong humoral and cellular responses compared to vaccination alone. However, the durability of these responses remains unknown [49].

Mucosal immune responses are important for early control of infectious disease and can generate long-term antigen-specific durable protection [50]. Despite this, currently approved SARS-CoV-2 vaccines do not provide adequate mucosal immunity, resulting in high rates of breakthrough infection and limited duration of vaccine-mediated protection [20,51]. We observed significantly elevated mucosal neutralizing antibodies against Wuhan, BA.2, and BA.5 in nasal swab samples from individuals who had received three vaccine doses and had a SARS-CoV-2 infection compared to those who received three vaccine doses but did not report having COVID-19. Investigators have evaluated the BAL (bronchoalveolar lavage) and serum samples from vaccinated donors with or without prior infection history for the presence of IgA and IgG titers of mucosal and circulating antibodies against the Spike, RBD, and N (nucleocapsid) protein of SARS-CoV-2 [52]. Compared to control individuals, vaccinees that had COVID-19 exhibited three-fold higher mucosal IgA titers against the spike and RBD protein. We observed undetectable or minimal IgA titers from triple-vaccinated individuals whereas triple-vaccinated and infected individuals showed 3–4-fold higher IgA response against RBD proteins of BA.1, BA.2, and BA.5 Omicron subvariants. Planas et. al., evaluated the neutralization response from sera and nasal swab samples following mRNA vaccination, up to 18 months, in individuals with or without Omicron infection and observed that antibody (IgA) levels against Omicron subvariants increased abruptly and remained elevated for at least 5–6 months in Omicron breakthrough individuals [53].

Additionally, increased nasal swab and serum cross-neutralization towards BA.2 and BA.5 further indicates that the breakthrough infections in our cohort likely originated from BA.2. A population-based survey conducted in Denmark reported high protection against BA.5 in triple-vaccinated individuals who had prior Omicron infection, especially BA.2, and described similar vaccine effectiveness against BA.2 and BA.5 [47]. These findings collectively support that durable mucosal immune response against SARS-CoV-2 develops only in individuals with hybrid immunity and cannot be induced by mRNA vaccinations alone. Various viral-vectored vaccines have been shown to improve mucosal immune responses against SARS-CoV-2 infection and are presently either in clinical trials or approved for human use [54,55,56].

## 5. Limitations of the Study

A limitation of this study is a lack of sequencing data to confirm the specific SARS-CoV-2 variants responsible for breakthrough infections in most individuals. However, we have confirmed the timing of infections and correlated them with infection waves and community wastewater strain prevalence during that period. We used authentic infectious virus neutralization assays, and NT_50_ or ID_50_ were calculated using normalized non-linear regression using GraphPad Prism software. Although neutralizing antibody titers remains a crucial predictor of protection against viral infection, no clear cutoff is available to define an NT_50_ threshold that indicates protective immunity. Additionally, another limitation is that we did not analyze the T-cell immunity although previous studies suggested that various T-cell epitopes remain preserved in Omicron spike protein after natural infection, which may have a greater impact in protection against severe diseases. Regardless of these limitations, our results showed that (1) three doses of current mRNA vaccine provide low neutralization against Omicron subvariants, especially against BA.5.; and (2) hybrid immunity provides better protection against re-infection, as vaccinated individuals with Omicron breakthrough infection exhibit greater neutralization breadth against other Omicron subvariants. Therefore, these findings warrant the development of COVID-19 vaccines targeting human respiratory mucosa.

## 6. Conclusions

These results emphasized that continuous surveillance of antibody levels in sera and nasal swabs is important for predicting vaccine efficacy and hybrid immunity against recently emerged and emerging SARS-CoV-2 variants. Our study highlights the importance of vaccine and infection-induced mucosal immunity and improves the knowledge, understanding, and breadth of immunological response against next-generation COVID-19 vaccines that utilize mucosal response.

## Figures and Tables

**Figure 1 vaccines-13-00559-f001:**
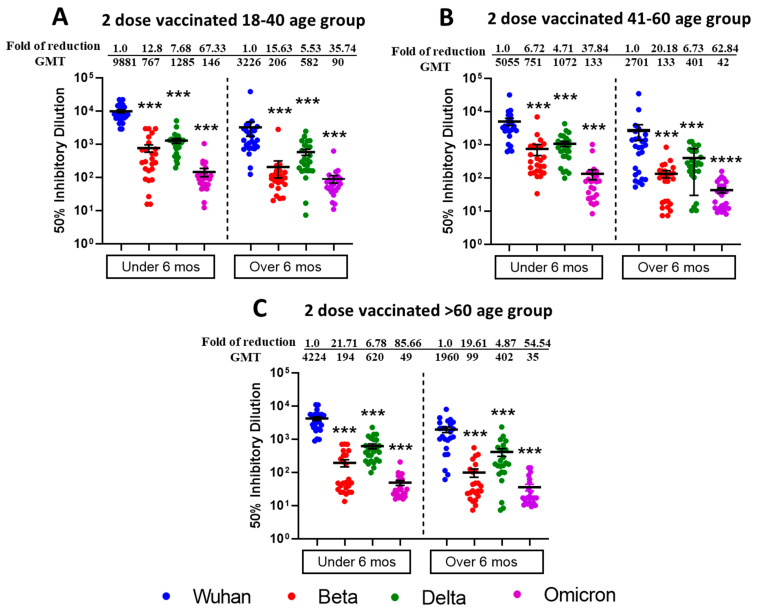
Omicron BA.1 is more resistant to neutralization by sera from individuals vaccinated with two doses of Moderna or Pfizer vaccine. 50% SARS-CoV-2 neutralization titers (VNT50) of sera collected from 18–40 age group (**A**), 41–60 age group (**B**), >60 age group (**C**), vaccine recipients collected within or after 6 months post two doses of either Pfizer or Moderna vaccine. Sera from participants were tested for neutralization against ancestral Wuhan Strain, Beta, Delta, and Omicron BA.1. following 1 h incubation with indicated SARS-CoV-2 VOCs before adding to VeroE6/TMPRSS2 cell monolayers. Each serum was tested in duplicate, and geometric mean 50% SARS-CoV-2 virus neutralization titers (VNT50) or 50% inhibitory dilutions were plotted. Values above the dots represent group GMTs and fold reduction in neutralization of Beta, Delta, and Omicron relative to Wild type ancestral Wuhan strain. Descriptive statistical analysis was performed using two-sided Friedman test with Dunn’s multiple comparison using 3 variables (age group, vaccine duration, and different virus types) and 2 comparisons (age group vs. virus type and vaccine duration vs. virus type). The asterisks (*** *p* < 0.001, **** *p* < 0.0001) above the dots represents significant fold reduction in neutralization compared with ancestral Wuhan strain. All statistical results have been shown in Appendix A.

**Figure 2 vaccines-13-00559-f002:**
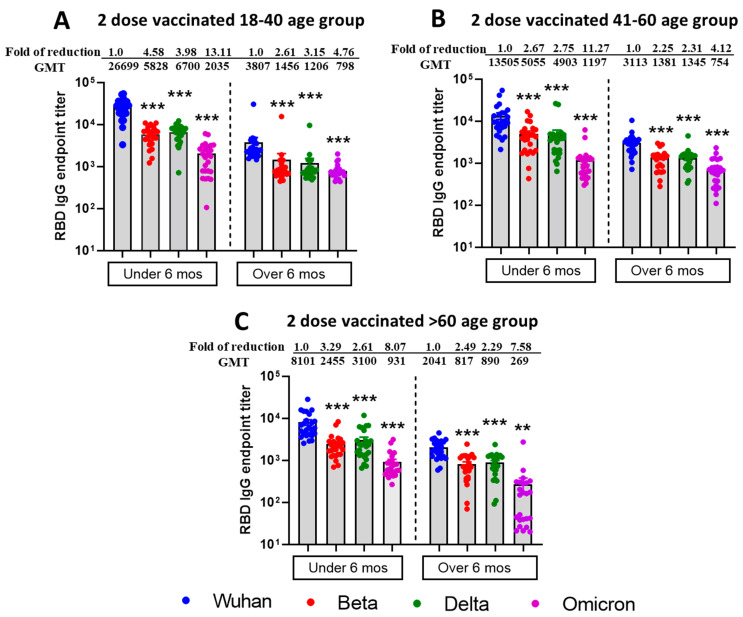
Reduced IgG binding titers against RBD protein of mutant SARS-CoV-2 VOCs from individuals vaccinated with the Moderna or Pfizer vaccine. ELISA IgG binding endpoint titers against different RBD of SARS-CoV-2 VOCs in sera collected from 18–40 age group (**A**), 41–60 age group (**B**), >60 age group (**C**), vaccine recipients collected within or after 6 months post two doses of either Pfizer or Moderna vaccine. Sera from participants were tested for binding interactions between RBD protein of different SARS-CoV-2 VOCs and antibody using endpoint titer ELISA. Serially diluted serum was added in duplicates to pre-coated RBD 96 well plates in duplicates, detected using anti-human IgG, and endpoint titers were calculated and plotted based on the cutoff value of naïve human sera (at 1:100 dilution). The fold reduction in neutralization of Beta, Delta, and Omicron relative to Wild type ancestral Wuhan strain. Descriptive statistical analysis was performed using two-sided Friedman test with Dunn’s multiple comparison using 3 variables (age group, vaccine duration, and virus type) and 2 comparisons (age group vs. virus type and vaccine duration vs. virus type). The asterisks (*** *p* < 0.001, ** *p* < 0.01) above the bars represent significant fold reduction in IgG binding titers to RBD protein compared with ancestral Wuhan strain. All statistical results have been shown in Appendix A.

**Figure 3 vaccines-13-00559-f003:**
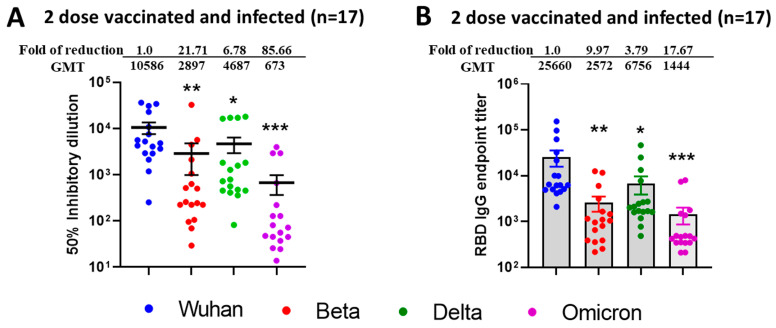
Breakthrough infection of previously vaccinated (2 doses) individuals induced broad neutralization of mutant SARS-CoV-2 VOCs including Omicron BA.1. 50% SARS-CoV-2 neutralization titers (**A**) and ELISA IgG binding endpoint titers (**B**) against different RBD of SARS-CoV-2 VOCs in sera collected from Modena or Pfizer Vaccinated and naturally COVID-19 infected individuals. (**A**) Values above the dots represents group GMTs, fold reduction, and significant difference in neutralization of Beta, Delta, and Omicron relative to Wild type ancestral Wuhan Strain by serum samples. (**B**) IgG endpoint titers were calculated and plotted based on the cutoff value against naïve human sera (at 1:100 dilution). Values above the bars indicate GMTs, fold reduction, and significant difference in IgG binding titers to RBD protein of Beta, Delta, and Omicron relative to Wild type ancestral Wuhan strain. For both experiments, the serum was tested in duplicates, and in each panel, data are mean from 2 independent experiments. A two-sided Friedman test with Dunn’s multiple comparisons was performed (* *p* < 0.05, ** *p* < 0.01, *** *p* < 0.001).

**Figure 4 vaccines-13-00559-f004:**
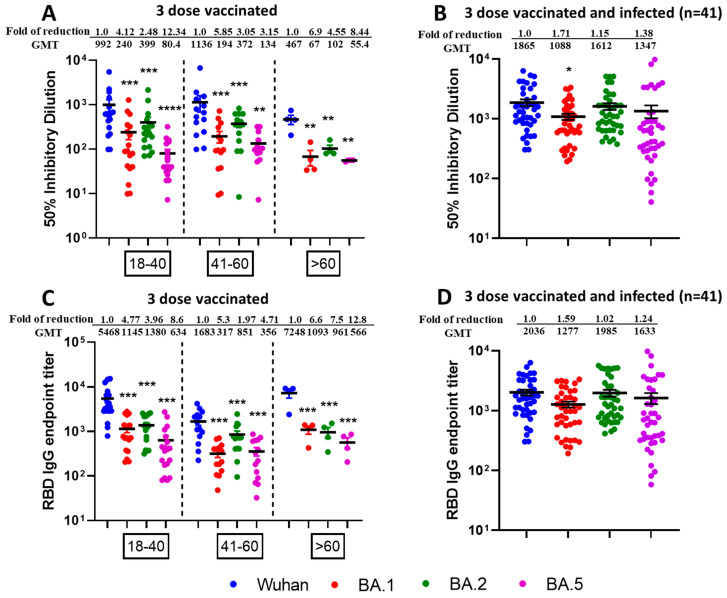
Omicron breakthrough infection of previously triple mRNA vaccinated individuals induces broad neutralization of Omicron sublineages. VNT50 titers of three-dose vaccinated (**A**) or three-dose vaccinated + infected sera samples (**B**), and IgG endpoint titers against RBD protein of three-dose vaccinated (**C**) or three-dose vaccinated + infected sera samples (**D**). Sera (n = 40) were collected from different age group—18–40 (n = 20), 41–60 (n = 15), and >60 (n = 5)—individuals who received three doses of vaccine (**A**) or Sera (n = 41) were collected from different age group—18–40 (n = 14), 41–60 (n = 21) and >60 (n = 6)—individuals who received three doses of vaccine and later were infected with Omicron (**B**) COVID-19 (breakthrough infections) were subjected to live-virus neutralization against ancestral Wuhan strain and recently emerged Omicron sublineages BA.1, BA.2, and BA.5. The fold reduction, GMTs, and significant difference in virus neutralization to BA.1, BA.2, and BA.5 relative to ancestral Wuhan strain were shown, and descriptive statistics using entire mixed model between each of the viral strains have been shown in Appendix A. For (**C**,**D**), IgG binding ELISA was conducted against RBD proteins of ancestral Wuhan strain or Omicron sublineages BA.1, BA.2, and BA.5. The serum was tested in duplicates and in each panel, data are mean from 2 independent experiments (* *p* < 0.05, ** *p* < 0.01, *** *p* < 0.001, **** *p* < 0.0001). The fold reduction, GMTs and significant difference in IgG binding titers against RBD proteins of BA.1, BA.2, and BA.5 relative to ancestral Wuhan strain were shown, and descriptive statistics using entire mixed model between each of the viral strains have been shown in Appendix A.

**Figure 5 vaccines-13-00559-f005:**
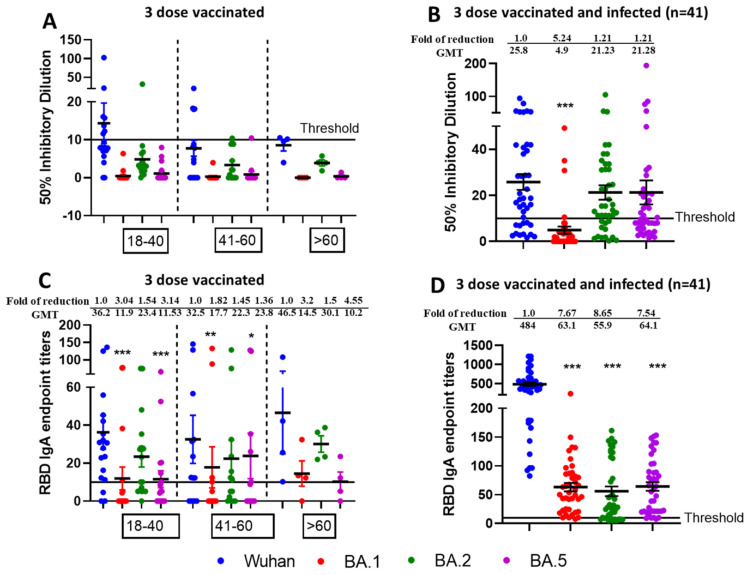
Generation of mucosal cross-neutralizing antibodies at nasal mucosa following vaccination and Omicron breakthrough infection resulting in hybrid immunity. Nasal swabs were collected after 6 months post third dose from vaccinated only (n = 40) and vaccinated + Omicron infected (n = 41) individuals and assessed for microneutralization and IgA binding assay against Wuhan, BA.1, BA.2, and BA.5. VNT50 titers of three dose vaccinated (**A**) or 3-dose vaccinated + infected nasal swabs (**B**) and IgA endpoint titers against RBD protein of three-dose vaccinated (**C**) or three dose-vaccinated + infected swab samples (**D**). * *p* < 0.05, ** *p* < 0.01, *** *p* < 0.001.

**Figure 6 vaccines-13-00559-f006:**
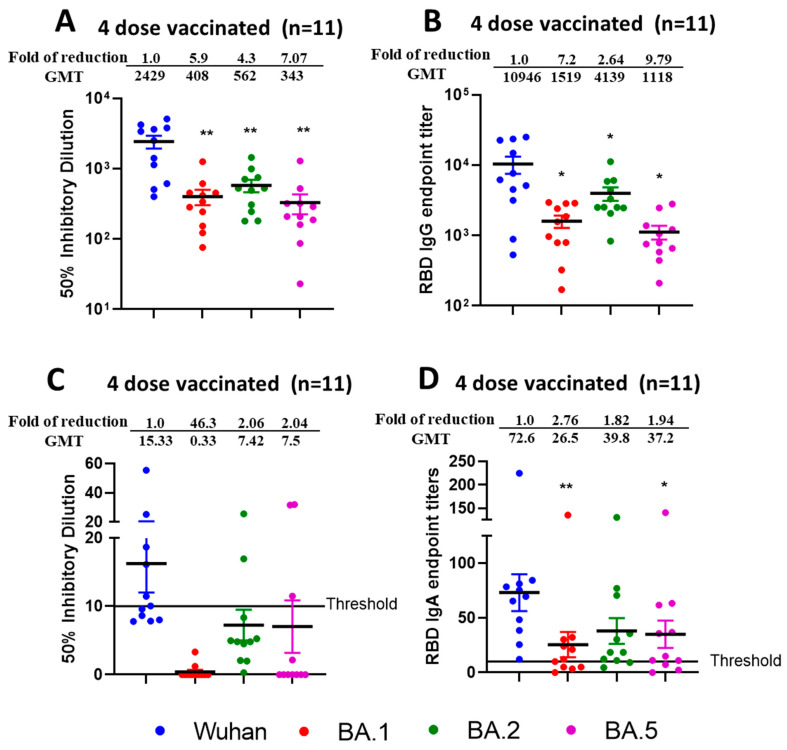
Reduced efficacy of neutralization against Omicron sublineages from sera and nasal swabs samples from quadruple mRNA vaccinated (double boosted) individuals. VNT50 titers from sera (**A**) and nasal swab (**C**); IgG endpoint titers against RBD (**B**) from sera and IgA endpoint titers against RBD from nasal swabs (**D**) of four dose vaccinated individuals (n = 11). Sera and nasal swabs were collected from >60 aged individuals who received four doses of vaccine and subjected to live-virus neutralization and RBD binding ELISA against ancestral Wuhan strain and recently emerged Omicron sublineages BA.1, BA.2, and BA.5. The fold reduction, GMTs and significant difference in virus neutralization to BA.1, BA.2, and BA.5 relative to ancestral Wuhan strain were shown from sera (**A**) and nasal swabs (**C**). The fold reduction, GMTs and significant difference in IgG from sera (**B**) and IgA from nasal swabs (**D**) binding titers against RBD proteins of BA.1, BA.2, and BA.5 were shown relative to ancestral Wuhan strain. The serum and swab samples were tested in duplicates and in each panel, data are mean from 2 independent experiments. (* *p* < 0.05, ** *p* < 0.01).

**Table 1 vaccines-13-00559-t001:** Mean, standard deviation (in parenthesis), and *p* value difference in terms of ID50 between vaccinated and infected (breakthrough) versus vaccinated only (18–40 age group) for different SARS-CoV-2 VOCs.

	Vaccinated and Infected (n = 17)	Vaccinated Only (n = 25)	*p*-Value
Wuhan	10,587 (12,245)	6554 (7292)	0.212
Beta	2897 (7872)	487 (829)	0.017
Delta	4688 (7188)	934 (912)	0.016
Omicron.BA.1	674 (1275)	118 (166)	0.097

**Table 2 vaccines-13-00559-t002:** Mean, standard deviation (in parenthesis), and *p* value difference for IgG endpoint titers between vaccinated and infected (breakthrough) versus vaccinated only (18–40 age group) for different SARS-CoV-2 VOCs.

	Infected	Vaccinated	*p*-Value
Wuhan	25,660 (41,375)	15,253 (14,933)	0.383
Beta	2573 (3848)	3642 (3586)	0.054
Delta	6757 (11,896)	3954 (3561)	0.522
Omicron.BA.1	1444 (2393)	1417 (1291)	0.091

**Table 3 vaccines-13-00559-t003:** Mean, standard deviation (in parenthesis), and *p* value in terms of ID50 between vaccinated and infected (breakthrough) versus vaccinated individuals for different SARS-CoV-2 VOCs.

	Infected	Vaccinated	*p*-Value
Wuhan	1861 (1517)	1303 (1571)	0.004
Omicron.BA.1	1032 (771)	248 (296)	<0.001
Omicron.BA.2	1522 (1211)	401 (405)	<0.001
Omicron.BA.5	1337 (2092)	152 (206)	<0.001

**Table 4 vaccines-13-00559-t004:** Mean, standard deviation (in parenthesis) and *p* value difference in terms of IgG endpoint titers between vaccinated and infected (breakthrough) versus vaccinated individuals for different SARS-CoV-2 VOCs.

	Infected	Vaccinated	*p*-Value
Wuhan	4910 (3738)	5662 (6246)	0.575
Omicron.BA.1	2963 (2922)	970 (875)	<0.001
Omicron.BA.2	5283 (3708)	1783 (1954)	<0.001
Omicron.BA.5	2466 (2011)	649 (686)	<0.001

**Table 5 vaccines-13-00559-t005:** Nasal swab mean ID50 value, standard deviation (in parenthesis), and *p* value for single boosted and infected versus boosted only individuals for SARS-COV-2 VOCs.

	Infected	Vaccinated	*p*-Value
Wuhan	26 (22)	12 (16)	0.003
Omicron.BA.1	5 (10)	0 (1)	<0.001
Omicron.BA.2	21 (20)	5 (6)	<0.001
Omicron.BA.5	21 (33)	2 (7)	<0.001

**Table 6 vaccines-13-00559-t006:** Nasal swab respiratory IgA mean, standard deviation (in parenthesis), and *p* value for single-boosted and infected versus boosted-only individuals for SARS-COV-2 VOCs.

	Infected	Vaccinated	*p*-Value
Wuhan	484 (314)	44 (47)	<0.001
Omicron.BA.1	63 (47)	17 (33)	<0.001
Omicron.BA.2	56 (54)	27 (31)	0.002
Omicron.BA.5	64 (47)	21 (35)	<0.001

## Data Availability

The data presented in this study are available upon request from the corresponding author.

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
