# Peer review of "Mucosal and Serum Neutralization Immune Responses Elicited by COVID-19 mRNA Vaccination in Vaccinated and Breakthrough-Infection Individuals: A Longitudinal Study from Louisville Cohort"

_vaccines, 2025, doi:10.3390/vaccines13060559_

Round 1
Reviewer 1 Report
Comments and Suggestions for Authors
This manuscript mainly investigated the mucosal and serum immune responses elicited by COVID-19 mRNA vaccination in vaccinated and breakthrough infection individuals in a longitudinal Louisville cohort. There are numerous major and minor concerns regarding this manuscript.
- In lines 361-362, sera samples were collected 6 months following third vaccine dose from different age groupsindividuals viz.18-40 (n=20), 41-60 (n=15), and >60 (n=5). The sample sizes across these groups vary considerably (only 5 individuals in the >60 group in Cohort II). Does this imbalance affect the statistical validity of the results? Has any consideration been given to adjusting the group sizes or increasing the sample size to enhance the robustness of the findings?
- The study categorizes participants into age groups (18-40, 41-60, and >60), but it remains unclear whether underlying health conditions or comorbidities in these groups might have influenced immune responses. Were these potential confounding factors excluded during cohort selection, or if not, how were they controlled for or accounted for in the statistical analysis? The manuscript mentions the collection of health data through “self-reported information”, but does not specify whether participants with specific comorbidities were excluded or whether these variables were adjusted for in the statistical model. Further clarification of the methodology is needed.
- Could the authors provide further justification for the use of different inclusion criteria across Cohort 1 (two doses of vaccine) and Cohort 2 (three doses of vaccine or breakthrough infection)? Was this design intentional, and how might it influence the interpretation of the results?
- When comparing results across cohorts (e.g., vaccine dose effects), were variables such as age and health status statistically adjusted for?If these factors were not adjusted, could you discuss potential biases introduced, such as whether the immune boost observed in the breakthrough infection group might be partially attributable to a younger population?
- The study was conducted in a cohort predominantly consisting of Kentucky residents, likely with a predominance of white individuals. To what extent are the results applicable to other regions or racial/ethnic groups? Did the authors consider potential differences in immune responses across different regions or demographic groups? Does this limit the broader applicability of the findings?
- Elevated nasal mucosal IgA levels were observed following breakthrough infection, but the study only investigates the short-term (6 months) effects. Are there plans to track mucosal antibody dynamics over a longer duration? Furthermore, is the observed elevation in mucosal IgA directly associated with reduced viral load or lower transmission risk?
- In the discussion, the authors refer to the consistency of their findings with other studies, such as the decline in neutralizing antibodies at different time points. Have the authors compared their results with studies specifically examining the persistence of immune responses against Omicron variants following vaccination and breakthrough infections? Could these studies help explain the discrepancies in immune responses observed in this study?
- Figure 4 illustrates superior neutralizing activity against BA.2 compared to BA.1/BA.5 following three doses of the vaccine, but this difference is not addressed in the discussion. Could this difference be due to the conserved nature of the Spike protein in BA.2, or are there other possible explanations, such as the differential effects of mutation sites on antibody escape?
- In the limitations section,the authors mention that breakthrough infections were primarily caused by Omicron variants, but do not provide sequencing data to confirm this. If temporal extrapolation was used, how can the influence of other circulating mutant strains be excluded?
- In the limitations section, the authors mention the absence of T-cell immunity analysis. Given the crucial role of T-cell responses in protecting against severe disease, especially against mutant strains, does the study plan to incorporate T-cell immunity assessments in future investigations? Additionally, the study primarily focuses on antibody titers without investigating their functional properties. Would it be possible to include tests for antibody functions, such as ADCC (antibody-dependent cellular cytotoxicity), in future studies?
- The quality of the figures presented is suboptimal. It is recommended that the clarity and resolution of the figures be improved.
- In Figure 1, the symbol “****” appears, but the figure legend only mentions (***p<0.001). Please clarify and specify the exact meaning of these statistical symbols in the figure legend. The same adjustment should be applied to other figures.
- In Figure 4, some groups are labeled with “*”, some with “ns”, and others are not labeled. For consistency, it is recommended to either label “ns” uniformly or remove these “ns”labels across all figures. The same should apply to other figures.
Author Response
We appreciate your efforts and precious time and providing comments, future directions and positive feedback for the manuscript.

Reviewer 2 Report
Comments and Suggestions for Authors
This well-conducted and timely study provides valuable longitudinal data on the immune response to monovalent mRNA COVID-19 vaccination and breakthrough infections, stratified by age group. The authors employed robust serological and virological assays, including microneutralization and ELISA, to evaluate systemic and mucosal immunity. Including nasal IgA data adds novelty and relevance to the manuscript, particularly in the context of mucosal vaccine development.
The findings contribute meaningfully to understanding vaccine-induced immunity and its enhancement following natural infection, especially against Omicron subvariants. The manuscript is well-written and well-structured, and the data support the conclusions.
Minor Revisions Needed: The graphical presentation should be standardized. The size and layout of graphs should be uniform across the figures to ensure clarity and visual consistency.
Author Response
We appreciate your efforts and precious time for providing comments, future directions and positive feedback for the manuscript.

Reviewer 3 Report
Comments and Suggestions for Authors
This manuscript assesses cross-variant SARS CoV-2 neutralizing and antibody responses in various conditions : i) in two cohorts of patients, one possibly retrospective, the other prospective, ii) both in sera and nasal swabs (mucosal responses), iii) either after vaccination or vaccination and COVID-19 breakthrough. The methods are rigorous and reliable, the amount of work considerable, the results very much detailed.
Major comments:
This study mostly confirms, with an enormous amount of work, previously published results (line 534): diminished neutralization against Omicron strains, waning over time, absence of evidence of the efficiency of additional vaccine doses to obtain responses against VOCs, efficiency of breakthrough infection to enhance neutralization efficiency.
The investigation of mucosal immunity (line 514) is a strong point, but notably under-emphasized.
There are too many figures and various parts of the results and discussion are redundant.
Minor comments:
Line 186: what is the reason for the different incubation period for Omicron variants?
Line 190: Was the cytopathic effect modified in comparison with the variants from previous waves? Please describe the various CPEs: syncytia ?
Line 287: were significantly different?
Line 470: did the persons investigated in the second cohort receive a vaccine based on a modified strain (not based on the Wuhan original strain)?
Author Response

(The authors gave the same response as above.)

Round 2
Reviewer 3 Report
Comments and Suggestions for Authors
Thank you for all the answers you provided.